# Non-invasive, vagus nerve stimulation to reduce ileus after colorectal surgery: protocol for a feasibility trial with nested mechanistic studies

Stephen J Chapman ,[1] Maureen Naylor,[2] Carolyn J Czoski Murray,[3] Damian Tolan ,[4] Deborah D Stocken,[5] David G Jayne[1]

¹Leeds Institute of Medical Research at St. James's, University of Leeds, Leeds, UK
²West Riding of Yorkshire Ileostomy Association, Leeds, UK
³Leeds Institute of Health Sciences, University of Leeds, Leeds, UK
⁴Leeds Teaching Hospitals NHS Trust, Leeds, UK
⁵Leeds Institute of Clinical Trials Research, University of Leeds, Leeds, UK

**Correspondence to**
Stephen J Chapman;
s.chapman@leeds.ac.uk

## ABSTRACT

**Introduction** Ileus is a common and distressing condition characterised by gut dysfunction after surgery. While a number of interventions have aimed to curtail its impact on patients and healthcare systems, ileus is still an unmet challenge. Electrical stimulation of the vagus nerve is a promising new treatment due to its role in modulating the neuro-immune axis through a novel anti-inflammatory reflex. The protocol for a feasibility study of non-invasive vagus nerve stimulation (nVNS), and a programme of mechanistic and qualitative studies, is described.

**Methods and analysis** This is a participant-blinded, parallel-group, randomised, sham-controlled feasibility trial (IDEAL Stage 2b) of self-administered nVNS. One hundred forty patients planned for elective, minimally invasive, colorectal surgery will be randomised to four schedules of nVNS before and after surgery. Feasibility outcomes include assessments of recruitment and attrition, adequacy of blinding and compliance to the intervention. Clinical outcomes include bowel function and length of hospital stay. A series of mechanistic substudies exploring the impact of nVNS on inflammation and bowel motility will inform the design of the final stimulation schedule. Semistructured interviews with participants will explore experiences and perceptions of the intervention, while interviews with patients who decline participation will explore barriers to recruitment.

**Ethics and dissemination** The protocol has been approved by the Tyne and Wear South National Health Service (NHS) Research Ethics Committee (19/NE/0217) on 2 July 2019. Feasibility, mechanistic and qualitative findings will be disseminated to national and international partners through peer-reviewed publications, academic conferences, social media channels and stakeholder engagement activities. The findings will build a case for or against progression to a definitive randomised assessment as well as informing key elements of study design.

**Trial registration number** ISRCTN62033341.

## Strengths and limitations of this study

► Feasibility outcomes will support the development of a definitive, randomised controlled trial, including considerations of recruitment, device compliance and sham blinding.

► Qualitative outcomes will explore possible barriers to participation and compliance with the self-administered device before and after colorectal surgery.

► The study will not explore the most appropriate outcome to be used in a definitive randomised controlled trial, but other related work done in parallel will fill this need.

## INTRODUCTION

Ileus is a distressing condition that occurs in 10%–20% of patients undergoing major abdominal surgery.[1] It is characterised by painful abdominal distension, persistent vomiting and delayed bowel function. This usually resolves 2–4 days after surgery, but in some cases, it may persist in excess of 10 days.[2] The impact of ileus is extensive. For patients, it increases the length of hospital stay and increases the risk of serious postoperative complications such as pneumonia, surgical site infection and venous thromboembolic events.[3] For hospitals, it is economically burdensome and is associated with an increase of 71% in healthcare costs.[4] Once considered to be a normal consequence of surgical recovery, the Association of Coloproctology of Great Britain and Ireland now considers ileus to be a research priority.[5]

In the last 20 years, a number of interventions to prevent ileus and its clinical sequelae have been explored.[6] Strategies to encourage early or sham feeding (chewing gum) have been studied extensively, but evidence for their effectiveness remains contentious.[7 8] New approaches to perioperative management (such as intravenous lidocaine) have been explored, but uncertainty relating to efficacy, dose and timing persists.[9] Novel drugs such as mu-opioid receptor antagonists have shown promising results, but economic and regulatory barriers

BMJ

have limited their uptake beyond North America. The consensus of existing literature suggests that minimally invasive surgery and protocol-driven recovery offer the best chances of preventing ileus.[10] Uncertainty exists over the pathophysiology of ileus, however, which is problematic for designing clinical interventions that are justified by strong scientific evidence.

Stimulation of the vagus nerve may represent a novel and evidence-based approach to reduce ileus after surgery. In preclinical models, electrical stimulation of the vagus nerve accelerates the recovery of gastrointestinal transit. This is facilitated through vagal afferents targeting the hypothalamic–pituitary–adrenal axis and vagal efferents targeting the cholinergic anti-inflammatory pathway, which attenuate inflammatory-mediated gut dysfunction.[11–13] Early clinical studies have suggested that this is directly translatable to humans undergoing surgery.[14 15] Challenges exist, however, as to how vagus nerve stimulation can be done safely, how its efficacy can be optimised and how patient acceptability can be ensured. A protocol for a randomised feasibility trial of non-invasive vagus nerve stimulation (nVNS) to reduce ileus, as well as a series of mechanistic and qualitative studies, is described.

## METHODS
### Ethics & governance
Ethical approval for the study was obtained from North East—Tyne and Wear South Research Ethics Committee (19/NE/0217) on 2 July 2019. The study was prospectively registered on the ISRCTN registry (http://www.isrctn.com/ISRCTN62033341) on 11 October 2019 prior to the start of recruitment (online supplemental file 1). The study devices will be provided through the

manufacturer's (electroCore) Investigator Initiated Trial Programme. The present manuscript is reported according to the Standard Protocol Items: Recommendations For Interventional Trials 2013 Checklist (online supplemental file 2).[16]

### Aims and objectives
The aim of this study is to determine the feasibility of performing a multicentre, phase III randomised controlled trial (RCT) of nVNS to reduce ileus after colorectal surgery. The following objectives will explore the case for progression to a phase III trial:
► To estimate the proportion of patients screened who are eligible for approach.
► To estimate the number of approached patients who consent to randomisation.
► To assess the adequacy of participant blinding to the allocated intervention.
► To assess participant compliance to the intervention schedule.
► To explore the safety of the intervention schedule.
► To estimate the rate of missing outcome data, rate of withdrawal and loss to follow-up.

### Study design
This study is a participant-blinded, parallel-group, randomised, sham-controlled feasibility trial (IDEAL Stage 2b) of nVNS to reduce ileus after colorectal surgery (figure 1). One hundred and forty participants will be recruited across two study sites. Participants will be randomised to four arms (1:1:1:1) according to the following stimulation schedules:
► Preoperative stimulation and postoperative stimulation.

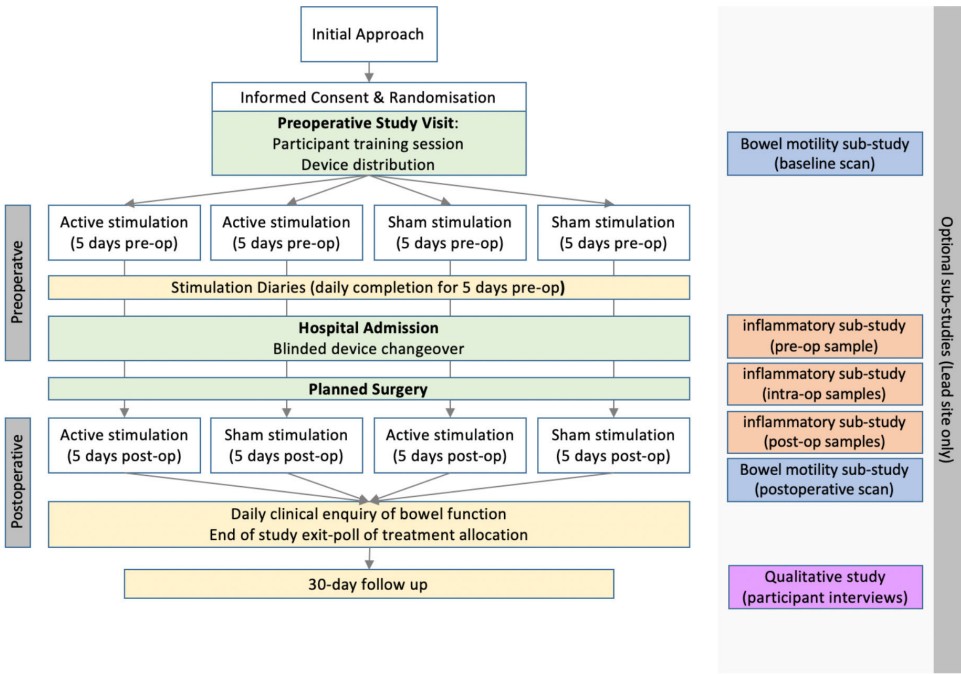

**Figure 1** Study flow chart.

**Table 1** Eligible procedures

| Right-sided resection | Left-sided resection |
|---|---|
| Ileocolic/caecal resection | Extended left hemicolectomy |
| Right hemicolectomy | Left hemicolectomy |
| Extended right hemicolectomy | Sigmoid colectomy |
| Transverse colectomy | Rectosigmoid colectomy |
| **Miscellaneous (side grouped accordingly)** | Anterior resection |
| Other segmental colonic resection | |

► Preoperative sham and postoperative stimulation.
► Preoperative stimulation and postoperative sham.
► Preoperative sham and postoperative sham.

The estimated trial duration is 24 months, and all participants will be followed up for 30 days after the date of surgery.

## Study setting

The intervention will take place within the community (preoperative stimulation) and in hospital (postoperative stimulation). All participants will undergo surgery at one of two large teaching hospitals (St James's University Hospital, Leeds, and Bradford Royal Infirmary, Bradford). Both provide elective colorectal cancer services including laparoscopic surgery within a programme of enhanced recovery.

## Eligibility criteria

To be considered eligible for the trial, patients must satisfy all of the following inclusion criteria:
► Aged ≥18 years.
► Able to provide written informed consent.
► Planned to undergo elective, minimally invasive (laparoscopic or robotic) surgery with a planned anastomosis/no routine plans for a diverting stoma, including one of the procedures listed in table 1.

Presence of any of the following exclusion criteria will preclude participation in the trial:
► Severe cardiac disease (myocardial infarction in the last 12 months, heart failure with New York Heart Association Scale ≥3, second-degree or third-degree atrioventricular block, permanent atrial fibrillation/flutter or previous ventricular tachycardia/fibrillation).
► Seizures or recurrent episodes of syncope (>1) in the last 5 years.
► Previous transient ischaemic attack or cerebral vascular accident.
► Previous vagotomy at any anatomical location.
► Confirmed diagnosis of chronic gastrointestinal inflammatory condition.
► Confirmed diagnosis of neuroendocrine tumour.
► Existing gastrointestinal stoma.
► Implanted electrical stimulator device (such as cardiac pacemaker or defibrillator).

► Structural abnormality of the neck anatomy that may impact on use of the device.
► Belonging to a vulnerable population (such as those lacking capacity and prisoners).
► Pregnant, nursing or thinking of becoming pregnant during the study period.

Participants will not be eligible for entry into other interventional studies that aim to optimise surgical recovery or for which the intervention may impact on bowel function.

## Intervention

nVNS will be self-administered by participants using the gammaCore device (electroCore, New Jersey, USA). This provides a transcutaneous electrical stimulation comprising of 5×5 KHz sine wave complexes delivered over 1 ms and repeated at 25 Hz (maximum voltage: 24 V; maximum output current: 60 mA). The surface landmark of the carotid pulse is used to guide positioning of the device over the cervical vagus nerve. Participants will self-administer the device twice daily for 5 days before and after surgery, with each administration lasting for 2 min on either side of the neck. Participants will be instructed to adjust the stimulation setting to the maximum tolerated level. A sham device identical in appearance will be used as a control. This produces a subtherapeutic stimulation (0.1 Hz biphasic direct current) while providing a perceptible stimulation sensation.

## Study outcome measures

The following outcomes will assess the feasibility of a phase III RCT of nVNS to reduce ileus:
► Proportion of eligible patients identified from screening logs.
► Number of eligible patients randomised (per month) and reasons for approach failure.
► Adequacy of participant blinding according to a blinding poll (Bang Blinding Index).[17]
► Compliance to the preoperative and postoperative stimulation schedules according to participant-reported diaries (maximum of 10 stimulations before and after surgery).
► Incidence of complications and serious complications within 30 days.
► Proportion of missing clinical outcome data, withdrawals and loss to follow-up.

Progression will be determined using the parameters outlined in table 2. 'Progress' outcomes will build a case for progression to a phase III trial. 'Adjust' outcomes will build a case for progression after appropriate amendments have been made. 'Stop' outcomes will contraindicate a case for progression, unless modifiable barriers are identified and further assessments of feasibility are undertaken. The predominance of such outcomes will be considered by the research team in the decision-making process.

Clinical data will be collected to explore variability in candidate measures including number of postoperative days to first flatus, first stool, first oral intake, hospital

**Table 2** Progression criteria

| Criteria | Stop | Adjust | Progress |
|---|---|---|---|
| Proportion of eligible patients identified from screening logs | <10% | 10%–20% | >20% |
| Number of eligible patients randomised over 24 months (site: SJUH) | ≤2 per month | 3–4 per month | ≥5 per month |
| Number of eligible patients randomised over 18 months (site: BRI) | <1 per month | 1–2 per month | ≥3 per month |
| Adequacy of participant blinding (according to Bang Blinding Index) | Index <−0.5 Or index >0.5 | Index −0.2 to −0.5 Or index 0.2–0.5 | Index 0 to −0.19 Or index 0–0.19 |
| Average compliance to the study treatment schedule | <10/20 stimulations across 10 days | 10–15/20 stimulations across 10 days | ≥16/20 stimulations across 10 days |
| Proportion of missing clinical outcome data | >40% | 15%–30% | <15% |
| Proportion of randomised patients lost to follow-up | >40% | 15%–30% | <15% |
| Incidence of complications or serious complications | >20% increase in complications | 5%–20% increase in complications | <5% increase in complications |

BRI, Bradford Royal Infirmary; SJUH, St James's University Hospital.

discharge and GI-2 (composite outcome of passage of stool and oral tolerance).[18]

## Participant timeline

The timing of study procedures, clinical assessments and collection points are summarised in table 3. Participants will attend a preintervention study visit, where a trained investigator will provide an interactive tutorial on how to self-administer the device. This will provide an opportunity to practice self-administration under direct supervision. Allocation of study devices will take place at two timepoints. The first will take place at the pre-intervention study visit, where devices will be provided for use in the community. On admission to hospital, these will be collected and switched with the postoperative device. Throughout the stimulation schedule, participants will record their compliance to the device schedule using stimulation diaries. Gastrointestinal function will be recorded daily, as reported by patients. On the last stimulation day, participants will complete a poll to explore the performance of blinding before and after surgery. Follow-up data (including the occurrence of complications and serious complications) will be collected 30 days after surgery to capture events, which may occur during the index hospital admission or after discharge.

## Sample size

As a feasibility study, no formal power calculation has been performed. In this four-arm parallel-group design, 35 participants per arm will allow robust estimates of variability in continuous clinical outcomes, and 70 patients randomised to pre-surgery and post-surgery groups will allow robust estimates of recruitment parameters.[19] The total sample size of 140 allows robust exploration of both clinical and recruitment-related objectives.

## Recruitment

Participants will be approached with written information in the outpatient departments of St James's University Hospital and Bradford Royal Infirmary. Research staff will be available to discuss the study over the phone, as well as opportunities to meet in person during patients' routine schedule of pre-assessment and anaesthetic review appointments. Consent for participation in the study will be gathered using the approved consent form (online supplemental file 3).

## Randomisation and allocation

Following confirmation of written informed consent, participants will be randomised to one of four intervention arms. The randomisation list will be prepared by an independent statistician using block randomisation with a variable block size. This will be stratified for type of colorectal resection (right-sided and left-sided) and recruitment centre. Randomisation will be facilitated centrally using an online 24-hour service and will be performed by study investigators.

## Blinding

Participants will be blinded to the study intervention before and after surgery. This is made possible through the use of a sham stimulator device, which is identical in appearance, weight and user interface, as well as being designed to provide a perceptible sub-therapeutic stimulation. All devices will be switched irrespective of whether the type of device (active or sham) remains constant throughout the pre-operative and post-operative schedules. Investigators and clinical outcome assessors will be unblinded to the intervention. Since outcomes relating to gastrointestinal function are patient-reported, the risk of assessor bias is considered to be low.

**Table 3** Schedule of events

| Events | | Baseline | Preop study visit | 5 days before surgery | Day of surgery (preop) | Day of surgery (intraop) | POD 1 | POD 2 | POD 3 | POD 4 | POD 5 | POD 6–10 | After discharge | 30-day phone F/U |
|---|---|---|---|---|---|---|---|---|---|---|---|---|---|---|
| Study procedures | Trial consent | ✓ | | | | | | | | | | | | |
| | Device training | | ✓ | | | | | | | | | | | |
| | Device distribution | | ✓ | | | | | | | | | | | |
| | Self-administered stimulation | | | ✓ | | | ✓ | ✓ | ✓ | ✓ | ✓ | | | |
| | Stimulation diary | | | ✓ | | | ✓ | ✓ | ✓ | ✓ | ✓ | | | |
| | Blinding poll | | | | ✓ | | | | | | ✓ | | | |
| Data collection | Eligibility CRF | ✓ | | | | | | | | | | | | |
| | Baseline CRF | ✓ | | | | | | | | | | | | |
| | Operative CRF | | | | | ✓ | | | | | | | | |
| | Gastrointestinal function CRF | | | | | | ✓ | ✓ | ✓ | ✓ | ✓ | ✓ | | |
| | Follow-up CRF | | | | | | | | | | | | | ✓ |
| Substudies* | Optional substudy consents | ✓ | | | | | | | | | | | | |
| | Inflammatory response study Blood only: ✓; blood and fluid: ✓✓ | | | | ✓ | ✓✓ | ✓ | ✓ | | | | | | |
| | Small bowel motility study (MRI) | | ✓ | | | | | ✓ | | | | | | |
| | Qualitative substudy (participant or non-participant interview) | | | | | | | | | | | | ✓ | |

*Substudies are optional and undertaken at the lead site only.
CRF, case report form; F/U, follow-up; POD, postoperative day.

## Data collection and management

All trial data will be prospectively entered onto case report forms by dedicated research staff (a clinical research fellow and research nurses). Physical data will be stored in a restricted-access research unit, accessible only to the research team. All electronic data will be stored in pseudo-anonymised format at the University of Leeds according to sponsor requirements. As a feasibility study, a data monitoring committee is not planned.

## Statistical analysis plan

All analyses will follow a pre-determined statistical analysis plan[20] and will be overseen by a qualified medical statistician. Recruitment, baseline characteristics, compliance and outcome data will be presented descriptively as rates (categorical) and means (continuous) with 95% CIs. No formal statistical hypothesis testing across randomised groups will be conducted. Groups may be collapsed to pre-surgery and post-surgery stimulus versus sham and reported descriptively. Blinding will be analysed according to the Bang Blinding Index, with an index of >0.2 representing unblinding, –0.2 to 0.2 representing random guesses and <–0.2 representing opposite guessing (also considered a source of unblinding).[17] All analyses will be performed for intention-to-treat (treatment as randomised) and per-protocol populations. Possible reasons for exclusion from the per-protocol population include non-compliance to the device schedule (defined as any deviation from the 5-day preoperative and postoperative stimulation schedule), conversion from laparoscopic to open surgery, formation of unplanned stoma, return to theatre prior to the return of bowel function and randomisation errors. The trial will be reported according to the Consolidated Standards of Reporting Trials Checklist.[21]

## Patient and public involvement

As a self-administered device, patient and public involvement is essential at each stage of the study. The design of this feasibility trial was informed through a patient focus group in collaboration with a study-specific patient representative. A patient advisory group comprising of 4–6 patients with experience of the intervention will be convened on six occasions across 3 years (including 1 year prior to study recruitment and 2 years during recruitment). The programme of discussion will be determined by the group but will include issues of patient approach, intervention burden, device training and design of study materials.

## Dissemination

Feasibility, mechanistic and qualitative findings will be disseminated to professional and patient groups via peer-reviewed publications, academic conferences, social media channels and stakeholder engagement activities.

## Trial substudies

Participants will be invited to take part in the following optional substudies.

---

> **Box 1  Substudy additional exclusion criteria**
>
> **Inflammatory response substudy**
> ► Regular use of non-steroidal anti-inflammatory drugs in the preceding 7 days.
> ► Medical condition requiring oral or injectable steroids.
> ► Neoadjuvant chemoradiotherapy within 12 months prior to surgery.
>
> **Bowel motility substudy**
> ► Indwelling pacemaker or cardiac defibrillator device.
> ► Non-magnetic resonance compatible metallic implants, prostheses, neurosurgical clips, indwelling stimulator or pump devices, which would preclude safe MRI.
> ► Claustrophobia.
> ► Metallic foreign bodies in the eyes.

### Inflammatory response substudy

The inflammatory response substudy will explore the effect of nVNS on measures of systemic and peritoneal inflammation. This will explore differences in efficacy between randomised arms, which will assist with the design of the final intervention. Recruitment to this substudy will take place at the lead site only (St James's University Hospital). Additional eligibility criteria will apply, as outlined in box 1. Venous blood draws will be performed at baseline, 2 hours, 24 hours and 48 hours after the start of surgery. Peritoneal lavage fluid will also be collected at the start and end of surgery. Lavages will be performed by instilling 20 mL of sterile 0.9% sodium chloride onto the small bowel using a Foley catheter and bladder tip syringe, followed by collection of 5–10 mL of fluid after 30 s, as described previously.[22] Serum (systemic) and fluid (peritoneal) levels of tumour necrosis factor alpha, interleukin (IL) 1 beta and IL-6 will be analysed using ELISA.[23 24]

### Bowel motility substudy

The motility substudy will explore the effect of nVNS on global small bowel motility. Differences in efficacy between randomised groups will inform the design of the final intervention. Recruitment will take place at the lead site only. Additional eligibility criteria (box 1) will apply with a recruitment target of 10 participants per group (total sample: 40 participants). Magnetic resonance enterography will be performed at baseline and on the third postoperative day in the supine position under breath hold conditions while a series of motility acquisitions are performed. Oral preparation with a single glass of water will take place 30 min prior to the scan. These will be registered with dedicated software and a quantitative motility score generated to depict bowel wall motion.[25]

### Qualitative substudy

A qualitative substudy will explore participants' acceptability of the trial recruitment process, intervention, their experiences of self-administration and any challenges with device blinding. Recruitment will take place at the lead site with a target sample of five participants per group

---

(total sample: 20 participants). In addition, five patients who declined participation in the study will be recruited to explore barriers to study enrolment. A maximum variation approach to sampling will be used to ensure that a wide variety of perspectives are captured (accounting for age, sex, indication for surgery, and device compliance). Semi-structured interviews will be conducted after hospital discharge in hospital or in the community. Questions will be informed by a topic guide and will be recorded in preparation for transcription. A thematic framework approach will be used for analysis, comprising five key stages: (1) familiarisation with data, (2) identifying the thematic framework, (3) indexing, (4) charting and (5) mapping and interpreting.[26] The thematic analysis will be modified in light of new data, and a process of constant comparison will be used to examine across themes and cases. Qualitative data will be managed in NVivo V.11 (Melbourne, Australia).

## DISCUSSION

Although ileus is a common problem after surgery, limited progress has been made to curtail its impact on patients and healthcare systems in the last 20 years.[6] The most successful measures have been the widespread introduction of minimally invasive surgery and the development of evidence-based enhanced recovery protocols. These gains are likely facilitated through reductions in inflammatory-induced and opioid-induced intestinal smooth muscle dysfunction, leading to faster recovery of bowel function.[27 28] The use of vagus nerve stimulation to reduce ileus is a new concept, which draws on similar principles. Its efficacy has been demonstrated consistently in pre-clinical studies through activation of a cholinergic anti-inflammatory pathway, leading to a faster restoration of intestinal transit.[11] In keeping with this, intraoperative stimulation of the vagus nerve in humans has been shown to attenuate systemic markers of inflammation after major abdominal surgery.[14] Taken together, this suggests that the anti-inflammatory mechanism of vagus nerve stimulation is directly translatable to patients and may represent a novel approach to reduce ileus after surgery.

Most previous studies of vagus nerve stimulation have involved surgical access to the neck or abdomen. This is challenging for clinical translation because the procedure is invasive and technically demanding. Non-invasive stimulation over the cervical surface landmark aims to mitigate these challenges and presents a number of possible advantages. One of these is the opportunity for stimulation before and after surgery. This is attractive since some evidence suggests that activation of the cholinergic anti-inflammatory pathway (using prucalopride) is most effective when done pre-operatively.[29] Another is the opportunity for stimulation on multiple occasions rather than a single intra-operative episode, which expands the scope for intervention. In this study, a schedule of 5 days will be used as this represents the time during which most patients are expected to regain bowel function.[2] Finally,

stimulation can be self-administered by patients, which encourages active participation. Previous evidence has shown that active participation is a key motivator during recovery after surgery.[30]

In contrast, there are several possible challenges associated with studying and implementing nVNS in this setting. First, patient self-administration requires adequate skill, understanding and compliance, which may be difficult in the post-operative setting. A dedicated programme of training, including device positioning, aims to mitigate this. Second, the non-invasive approach to stimulation is less precise than conventional stimulation since contact with the vagus nerve cannot be confirmed objectively. This is mitigated through the use of a standard technique (stimulation over the cervical surface landmark), which has been shown to increase heart rate variability, reduce markers of systemic inflammation and generate vagal somatosensory evoked potentials (indicators of vagal stimulation) previously in healthy volunteers.[31–34] Finally, identifying an adequate sham intervention by which to compare nVNS is challenging. In previous healthy volunteer studies, nVNS has been shown to significantly reduce the expression of inflammatory cytokines from baseline but not when using the sham device.[32] It is possible that manual instrumentation of the device may be sufficient to elicit some vagal activity, but this is considered minimal and is comparable across both device types. Providing that these challenges can be addressed, nVNS may provide a practical approach to stimulating the vagus nerve in the pre-operative and post-operative setting.

The present study will inform the feasibility of a phase III RCT of nVNS by addressing key challenges in trial design. This includes compliance to the stimulation schedule in a complex clinical setting where unique barriers may exist. It also includes the adequacy of participant blinding. This is important in studies of electrical nerve stimulation where sham controls may be associated with statistical improvements in clinical outcomes.[35] Some issues of feasibility are beyond the scope of this study. The selection of outcomes for ileus is poorly standardised, with over 50 different outcomes identified from previous literature in the last 20 years.[36] To address this, a core outcome set for gastrointestinal recovery is under development by an international collaboration of clinicians and patients.[37] This will help to standardise reporting in the future. In addition, the feasibility of implementing nVNS into enhanced recovery after surgery protocols across diverse settings will remain unclear. The present study explores the feasibility of nVNS in two large teaching hospitals, but further consideration of barriers to implementation will be required.

**Contributors** SJC and DJ conceptualised the study, and all authors (SJC, MN, CJCM, DT, DDS and DJ) contributed to its design. SJC prepared the protocol draft, which was reviewed and approved by all authors (SJC, MN, CJCM, DT, DDS and DJ). The Clinical Trials Research Unit at University of Leeds oversees randomisation, and the University of Leeds is the study sponsor. DDS is the statistical guarantor, and DJ is the overall study guarantor.

**Funding** This study is funded by a National Institute for Health Research Doctoral Research Fellowship (SJC; DRF-2018-11-049). Study devices are provided through the manufacturer's (electroCore) Investigator Initiated Trial Programme.

**Competing interests** None declared.

**Patient consent for publication** Not required.

**Ethics approval** Research ethics approval was confirmed by the North East—Tyne and Wear South NHS Research Ethics Committee (19/NE/0217) on 2 July 2019. Written, informed consent to participate will be obtained from all participants.

**Provenance and peer review** Not commissioned; externally peer reviewed.

**ORCID iDs**
Stephen J Chapman http://orcid.org/0000-0003-2413-5690
Damian Tolan http://orcid.org/0000-0001-9895-9874

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
