## [Reviewer comments · BMJ Open]

ARTICLE DETAILS

TITLE (PROVISIONAL)	Non-invasive, vagus nerve stimulation to reduce ileus after colorectal surgery: Protocol for a feasibility trial with nested mechanistic studies
AUTHORS	Chapman, Stephen; Naylor, Maureen; Czoski Murray, Carolyn; Tolan, Damian; Stocken, Deborah; Jayne, David

VERSION 1 – REVIEW

REVIEWER	Gottfried Blackmore, Andres Stanford University, Medicine, Gastroenterology
REVIEW RETURNED	20-Nov-2020

GENERAL COMMENTS	November 20, 2020 bmjopen-2020-046313 In this study protocol, Chapman et al propose to assess the feasibility of an interventional trial using non-invasive vagal nerve stimulation (nVNS) to improve outcomes of post-operative ileus. The major strengths here are that a) the trial addresses both a knowledge gap and a medical need with broad implications for the results; b) it includes important secondary mechanistic studies of inflammation and bowel motility; c) the protocol is presented in a detailed manner. However, there are some limitations impacting the clarity of objectives and protocol outcomes of the study. It is unclear how this feasibility study protocol can be separated from its actual interventional nature as a clinical trial. Similarly, the wording of “IDEAL Stage 2b” was confusing to this reviewer, as the study is not described as exploratory in the text, nor does it refer to any surgical procedure investigation/variation. Rather, the study comes across as a formal randomized single-blinded interventional study, and a preliminary feasibility study for a phase-III RCT. Additionally, details about clinical outcomes and blinding of the intervention are lacking. As an already initiated trial, the Principal Investigator’s ability to modify/update the protocol is unclear. However, from a reviewer’s perspective, the following points are important to discuss/correct in a published/updated research protocol (and will likely be relevant in future presentation and discussion of the results).
--

Concerns:

1. What was the rationale for not selecting an a priori clinical outcome? By definition, an interventional trial requires a defined and measurable expected outcome. Per authors:

“ The study will not explore the most appropriate outcome to be used in a definitive randomised controlled trial, but other related work done in parallel will fill this need.”

There is no reference, trial number, or explanation for “other related work done in parallel”.

2. As a feasibility interventional study, there seems to be lacking a definition of a primary clinical endpoint (length of stay? time to first bowel movement?, pre-established definition of ileus and its rate of occurrence?); as well as its measurement, and feasibility cut-off criteria (as in Table 2), to determine if such a primary endpoint should be discarded/modified/kept in future larger RCTs. This important point is even discussed by the authors at the end of the manuscript.

3. Further, it is not clear how primary clinical endpoint(s) will be used to inform futility, which is a major ethical consideration in extending any study to a new Phase-III RCT; i.e. even with excellent enrollment, blinding, compliance, and record keeping in this feasibility protocol, a phase-III RCT would not proceed if there was no signal/evidence of clinical benefit. This is especially relevant when the authors have placed so much effort to assess the effects of placebo/sham and blinding in their study.

4. A power calculation is lacking, regardless of whether there is a primary clinical outcome or not. Given the extensive literature published on post-op ileus, the authors may be able to assess variance in the clinical outcome variables they are studying (time to flatus, time to bowel movement, time to first meal, etc.) and ascertain if they are powered to see a trend/difference with their intervention(s).

Furthermore, prior clinical trials at the designated study centers (St. James’s University Hospital and Bradford Royal Infirmary) may provide valuable information regarding the major feasibility outcomes proposed such as rates of enrollment, retention, compliance, and follow-up in interventional vs non-interventional clinical trials from the adult patient population visiting these centers.

5. Blinding: The authors state that “Given the objective clinical outcomes, the risk of assessor bias is considered low.” This is not entirely correct, particularly if the clinical outcome is based on patient recall (time to flatus or time to first bowel movement).

Assessors may indeed bias participants to modify their recall/responses in a scenario where the sham group is interrogated less frequently compared to the treatment group. This concern may be alleviated or worsened depending on the unit of measurement (days vs hours, respectively) to be selected for the clinical outcomes proposed.

6. Clinical outcome measurements: Several details need to be provided that are missing. Who will record the clinical events? It is stated that "All trial data will be entered on to electronic case report forms by research staff." Will this staff be in the post-operative care units to retrieve clinical outcomes? Or will these be retrospectively obtained from nursing charts? Reference study 28, for example, used a blinded research RN for daily clinical assessments.

7. Has the sham condition/settings on the gammaCore device been demonstrated to avoid VNS?

8. Will the gammaCore treatment protocol instruct subjects to increase voltage to maximum tolerated, and/or until there is contraction of the ipsilateral orbicularis oris muscle (as has been done in other gammaCore trials)? If a fixed voltage setting will be used, how will investigators ensure efficiency of nVNS on different neck anatomical subtypes (neck girth, length, subcutaneous or muscle mass, etc.)?

9. How will effective blinding take place at the time of device training? Particularly, if the treatment protocol requires subjects to increase the voltage (until their ipsilateral orbicularis oris muscle twitches), while presumably this will not occur with the sham device?

Minor:

1. What is meant by "progression to a phase-III trial"? It is not clear if this feasibility trial will be conducted and completed with 140 subjects, or if it is meant to continue enrollment beyond 140 subjects for a phase-III RCT, or if it will be stopped early based on the parameters in Table 2.

2. What units of measurement will be used for GI clinical outcomes (post-op days vs hours)? Will these be recorded objectively as to avoid patient recall bias? i.e. time to first flatus?

	3. Exclusion criteria: re vagotomy of any kind. Would expand this to procedures with potential to disrupt the integrity of major supra- and sub-diaphragmatic vagal branches, i.e. gastric fundoplication, partial gastrectomy, esophagectomy, etc. 4. Would vulnerable populations not be eligible for enrollment? Particularly when the pre-operative protocol is only 5 days, twice a day for 4 minutes. 5. What is the rationale for pre-operative therapy? How was the 5-day duration of therapy arrived at? Reference 28 provided in the discussion cites a paper in which prucalopride, a selective serotonin receptor agonist, but not VNS, improved ileus outcomes in patients. A reference showing that VNS, or nVNS, increases serotonin signaling in the gut would make this argument stronger. 6. Patient and public involvement. It is stated that a patient advisory group will be convened 6 times across 3 years. How will this be possible if the study proposed is only 2 years? 7. In secondary outcomes, Inflammatory response sub-study: peritoneal fluid analysis is not necessarily equivalent to “intestinal inflammation, given that the majority of immune/inflammatory cells of the gut are in the submucosa, where as the peritoneal cavity has a large influx of peritoneal mononuclear cells. Further, peritoneal fluid cytokines are likely originating from multiple sites including disrupted serosal surfaces at sites of bowel resection/anastomosis and peritoneal membranes at site of abdominal incisions. Perhaps the authors may consider “peritoneal inflammation” vs “intestinal” for clarity and accuracy. 8. SPIRIT checklist. Interventions, points 11b and 11c are missing. For example, subject adherence and compliance to nVNS in the pre-op period can be improved by having research staff check in on day 2-3 via telephone to assess compliance and clarify concerns/questions.
--	---

REVIEWER	O'Grady, Gregory University of Auckland
REVIEW RETURNED	23-Nov-2020

GENERAL COMMENTS	Thank you for the opportunity to review this article for BMJ Open titled “Non-invasive, vagus nerve stimulation to reduce ileus after colorectal surgery: Protocol for a feasibility trial with nested mechanistic studies”, authored by Chapman et al. from University of Leeds. This is a study protocol for a randomised, sham-controlled feasibility study of non-invasive vagus nerve stimulation in patients
--

	undergoing elective laparoscopic colorectal surgery. The authors are to be commended for undertaking such an ambitious body of work and applying a rigorous approach to the design and mechanistic evaluation of an eventual Stage 3 trial. Major comments:  1. Will patients planned to have a diverting stoma, for example to cover a high-risk rectal anastomosis, be eligible for inclusion in the study? 2. What evidence is available demonstrating that the 0.1Hz biphasic direct current used as a sham stimulation is sub-therapeutic for the purposes of nVNS? This statement should be referenced. 3. Have the authors considered how the crossover design may affect the assessment of blinding at the POD5 time point? I.e. Patients may be adequately blinded pre-operatively but then become unblinded during the post-operative period if the change between active/sham stimulation alters the perceptible stimulus of stimulation? It may be beneficial to assess blinding pre-operatively on the day of surgery as well as on POD5 – particularly given adequacy of blinding is a part of the progression criteria. 4. How will non-compliance be defined for the purposes of exclusion of patients from the intention-to-treat analysis? 5. Is there any anticipated bias in the MRE sub-study as to whether patients taking part in this will be able to tolerate sufficient oral intake for the enterography oral preparation? Minor comments:  1. “Systematic” in paragraph 3 on Page 15 should read “systemic”. 2. What data will be collected from patients at the 30-day point as part of the follow-up CRF? The clinical outcomes listed on Page 12 are likely to all be met either as an inpatient or very early following discharge. 3. Much of the evidence for VNS in abdominal surgery has been in the setting of foregut surgery. This may not be directly applicable to the colorectal setting where the hindgut does not receive direct vagal innervation, and this is a potential limitation of this approach which the authors may like to consider in the discussion. Good luck with the trial, we will be awaiting the results with great interest.
--	--

VERSION 1 – AUTHOR RESPONSE

Reviewer 1; Comment 1:

There are some limitations impacting the clarity of objectives and protocol outcomes of the study. It is unclear how this feasibility study protocol can be separated from its actual interventional nature as a clinical trial.

Thank you for this comment. We note a number of other specific comments relating to the feasibility design and objectives. These are addressed in further detail below.

We define a feasibility study according to the definition set out by the UK National Institute of Health Research (NIHR), found here:

<https://www.nihr.ac.uk/documents/additional-guidance-for-applicants-including-a-clinical-trial-pilot-study-or-feasibility-as-part-of-a-personal-award-application/11702>

In summary, feasibility studies do not evaluate the outcome of interest. This is left to the main study. They are used to answer prospectively identified feasibility questions, with the main aim of determining “can this study be done”. This is distinct from a pilot study, which may resemble the main study and may include an assessment of the primary outcome.

In the present manuscript, an outline of the feasibility objectives and relevant outcomes are described as appropriate.

Reviewer 1; Comment 2:

The wording of “IDEAL Stage 2b” was confusing to this reviewer, as the study is not described as exploratory in the text, nor does it refer to any surgical procedure investigation/variation”

Thank you for this comment. The IDEAL Framework describes stages through which new surgical interventions/innovations pass, from proof-of-concept to long-term surveillance.

http://www.ideal-collaboration.net/wp-content/uploads/2011/07/stages_of_surgical_innovation.jpg

We consider this to be a Stage 2b study, since proof-of-concept (Stage 1) and safety (Stage 2a) of nVNS in the setting of abdominal surgery have already been described. The aim of the present study is to explore the feasibility of a definitive randomised assessment (Stage 3).

Reviewer 1; Comment 3:

What was the rationale for not selecting an a priori clinical outcome? By definition, an interventional trial requires a defined and measurable expected outcome.

Thank you – please see response to Reviewer 1; Comment 1. As a feasibility study, the aim is not to evaluate the primary clinical outcome of interest. Instead, a series of feasibility objectives and outcomes are described to determine if a definitive study can be done.

Reviewer 1; Comment 4:

There is no reference, trial number, or explanation for “other related work done in parallel

Thank you, this is described in detail in the final paragraph of the Discussion, including citations to the appropriate work (references 35 and 36).

Reviewer 1; Comment 5:

As a feasibility interventional study, there seems to be lacking a definition of a primary clinical endpoint (length of stay? time to first bowel movement? pre-established definition of ileus and its rate of occurrence?); as well as its measurement, and feasibility cut-off criteria (as in Table 2), to determine if such a primary endpoint should be discarded/modified/kept in future larger RCTs. This important point is even discussed by the authors at the end of the manuscript.

Thank you. Please see response to Reviewer 1; Comment 1.

A selection of candidate primary outcomes (such as days to first flatus, stool, and oral tolerance) are collected for the purpose of exploring data variability and to inform a future sample size calculation. We agree that the final selection of clinical outcomes is an important consideration but this is beyond the scope of the present feasibility study. Instead, this is considered in other collaborative work done elsewhere (involving the development of a core outcome set). As noted, a description of this is provided in the Discussion.

Reviewer 1; Comment 6:

Further, it is not clear how primary clinical endpoint(s) will be used to inform futility, which is a major ethical consideration in extending any study to a new Phase-III RCT; i.e. even with excellent enrollment, blinding, compliance, and record keeping in this feasibility protocol, a phase-III RCT would not proceed if there was no signal/evidence of clinical benefit. This is especially relevant when the authors have placed so much effort to assess the effects of placebo/sham and blinding in their study.

Thank you for this comment. We agree that the issue of futility is important, but this is not the role of a feasibility study (please see response to Reviewer 1; Comment 1).

Reviewer 1; Comment 7:

A power calculation is lacking, regardless of whether there is a primary clinical outcome or not. Given the extensive literature published on post-op ileus, the authors may be able to assess variance in the clinical outcome variables they are studying (time to flatus, time to bowel movement, time to first meal, etc.) and ascertain if they are powered to see a trend/difference with their intervention(s). Furthermore, prior clinical trials at the designated study centers (St. James's University Hospital and Bradford Royal Infirmary) may provide valuable information regarding the major feasibility outcomes proposed such as rates of enrollment, retention, compliance, and follow-up in interventional vs non-interventional clinical trials from the adult patient population visiting these centers.

Thank you. We are guided by previous literature for determining the sample size of a feasibility study. Here, we have selected the guidance provided by Teare and colleagues (Reference 19) for estimating key design parameters (n=35 participants per group).

Reviewer 1; Comment 8:

Blinding: The authors state that "Given the objective clinical outcomes, the risk of assessor bias is considered low." This is not entirely correct, particularly if the clinical outcome is based on patient recall (time to flatus or time to first bowel movement). Assessors may indeed bias participants to modify their recall/responses in a scenario where the sham group is interrogated less frequently compared to the treatment group. This concern may be alleviated or worsened depending on the unit of measurement (days vs hours, respectively) to be selected for the clinical outcomes proposed.

Thank you for this comment. A standard approach to collecting clinical data is used for all participants. This includes completion of daily case report forms (Table 3: Schedule of Events) including data relating gastrointestinal function (i.e. passage of flatus, passage of stool, oral tolerance). Since these are reported wholly by patients, the risk of assessor bias is considered to be low. The following clarifications have been made:

— Blinding: Since outcomes relating to gastrointestinal function are patient-reported, the risk of assessor bias is considered to be low.

- Participant Timeline: Gastrointestinal function will be recorded daily, as reported by patients.

Reviewer 1; Comment 9:

Clinical outcome measurements: Several details need to be provided that are missing. Who will record the clinical events? It is stated that “All trial data will be entered on to electronic case report forms by research staff.” Will this staff be in the post-operative care units to retrieve clinical outcomes? Or will these be retrospectively obtained from nursing charts? Reference study 28, for example, used a blinded research RN for daily clinical assessments.

Thank you for seeking clarification on this. All data are collected prospectively by dedicated research staff (i.e. a clinical research fellow and research nurses). The following clarification has been made:

- Data Collection & Management: All trial data will be prospectively entered on to case report forms by dedicated research staff (a clinical research fellow and research nurses).

Reviewer 1; Comment 10:

Have the sham condition/settings on the gammaCore device been demonstrated to avoid VNS?

Thank you for this comment. Previous studies in healthy volunteers have demonstrated that active vagus nerve stimulation (using the GammaCore device) reduces the expression inflammatory cytokines from baseline (Reference 32). This is not observed when using the sham device (0.1Hz biphasic direct current). It is possible that manual instrumentation of the device may be sufficient to produce some vagal activity, but this is considered minimal and is comparable across both active and sham devices. The following has been added:

Discussion: There are several possible challenges associated with studying and implementing nVNS in this setting. Firstly, self-administration requires adequate skill, understanding, and compliance, which may be difficult in the post-operative setting. A dedicated programme of training, including device positioning and administration, aims to mitigate this. Secondly, the non-invasive approach to stimulation is less precise than conventional stimulation since contact with the vagus nerve cannot be confirmed objectively. This is mitigated through the use of a standard technique (stimulation over the cervical surface landmark), which has been shown to successfully increase heart rate variability, reduce markers of systemic inflammation, and generate vagal somatosensory evoked potentials (indicators of vagal stimulation) in healthy volunteers (31-34). Finally, identifying an adequate sham intervention by which to compare nVNS is challenging. In previous healthy volunteer studies, nVNS has been shown to significantly reduce the expression of inflammatory cytokines from baseline but not when using the sham device (32). It is possible that manual instrumentation of the device may be sufficient to elicit some vagal activity, but this is considered minimal and is comparable across both device types. Providing that these challenges are addressed, nVNS may provide a practical approach to stimulating the vagus nerve in the pre- and post-operative setting.

Reviewer 1; Comment 11:

Will the gammaCore treatment protocol instruct subjects to increase voltage to maximum tolerated, and/or until there is contraction of the ipsilateral orbicularis oris muscle (as has been done in other gammaCore trials)? If a fixed voltage setting will be used, how will investigators ensure efficiency of nVNS on different neck anatomical subtypes (neck girth, length, subcutaneous or muscle mass, etc.)?

Thank you. All participants are instructed to increase the stimulation voltage to the maximum tolerated setting. This is explained in the device training programme and participants are asked to record the setting using the device diary (Table 3: Schedule of Events). Contraction of the orbicularis oris muscle is not discussed as a means to guide positioning/setting of the device since this may lead to unblinding in participants who are allocated the sham device. The following clarification has been made:

— Intervention: Participants will self-administer the device twice daily for five days before and after surgery, with each administration lasting for 2 minutes on either side of the neck. Participants will be instructed to adjust the stimulation setting to the maximum tolerated level.

Reviewer 1; Comment 12:

How will effective blinding take place at the time of device training? Particularly, if the treatment protocol requires subjects to increase the voltage (until their ipsilateral orbicularis oris muscle twitches), while presumably this will not occur with the sham device?

Thank you. As per previous studies, participants are instructed to position the device over the surface landmark of the cervical vagus nerve (i.e. the carotid pulsation). Instead of increasing the stimulation setting to the point of eliciting muscle twitching, participants are instructed to increase the setting to the maximum tolerated level. Muscle twitching is described as a possible effect. The performance of blinding (and reasons for unblinding) will be explored using the blinding poll and during the qualitative sub-study (Page 14).

Reviewer 1; Comment 13:

What is meant by “progression to a phase-III trial”? It is not clear if this feasibility trial will be conducted and completed with 140 subjects, or if it is meant to continue enrollment beyond 140 subjects for a phase-III RCT, or if it will be stopped early based on the parameters in Table 2.

Thank you. The progression criteria outlined in Table 2 describe situations in which a definitive trial may be considered feasible or not feasible. These will inform the study conclusions by justifying or precluding the feasibility of a definitive trial in the future. Importantly, the study is not an internal pilot, which is characterised by a preliminary period of recruitment within the scope of a definitive trial.

Reviewer 1; Comment 14:

What units of measurement will be used for GI clinical outcomes (post-op days vs hours)? Will these be recorded objectively as to avoid patient recall bias? i.e. time to first flatus?

Thank you, this has been clarified as below:

— Study Outcome Measures: Clinical outcomes will be collected to explore variability in candidate measures including: number of postoperative days to first flatus, first stool, first oral intake, hospital discharge, and GI-2 (composite outcome of passage of stool and oral tolerance) (18).

Reviewer 1; Comment 15:

Exclusion criteria: re vagotomy of any kind. Would expand this to procedures with potential to disrupt the integrity of major supra- and sub-diaphragmatic vagal branches, i.e. gastric

fundoplication, partial gastrectomy, esophagectomy, etc.

Thank you – this is a good point. We are restricted in our ability to change the prospective methods since recruitment has already begun. We do not consider this scenario to be a frequent occurrence and the issue will be an interesting point of discussion in future reports.

Reviewer 1; Comment 16:

Would vulnerable populations not be eligible for enrollment? Particularly when the preoperative protocol is only 5 days, twice a day for 4 minutes.

Thank you. In this context, “vulnerable groups” relate to participants who are not be able to provide informed consent, such as due to lack of capacity or deprivation of liberty. Since the study involves an unsupervised/self-administered treatment, inclusion of these groups is not considered safe or appropriate for any duration.

Reviewer 1; Comment 17:

What is the rationale for pre-operative therapy? How was the 5-day duration of therapy arrived at? Reference 28 provided in the discussion cites a paper in which prucalopride, a selective serotonin receptor agonist, but not VNS, improved ileus outcomes in patients. A reference showing that VNS, or nVNS, increases serotonin signalling in the gut would make this argument stronger.

Thank you. In the noted citation (now amended to Reference 29) the use of prucalopride is shown to reduce ileus when administered before surgery but not when administered after. In other studies (such as References 14 & 15), VNS after surgery has demonstrated convincing signals of benefit. As described in the text, prucalopride and VNS share the same mechanistic pathway for reducing ileus (the cholinergic anti-inflammatory pathway). There is uncertainty as to whether pre- or post-operative stimulation is most efficacious and this will be explored in the inflammatory response sub-study (Page 13).

Previous evidence has shown that nVNS (using the GammaCore device) reduces the expression of inflammatory cytokines in healthy volunteers when administered once and three times daily (References 31 & 32). The selection of twice daily stimulation is considered to balance treatment efficacy with the practicality required for self-administration. In most patients, the return of gastrointestinal function is expected within five days of surgery and thus represents the most appropriate window for intervention. For consistency, the same period is selected for pre-operative stimulation. The following clarification has been made:

— Discussion: Stimulation can be performed before and after surgery, which is attractive since some evidence suggests that activation of the cholinergic anti-inflammatory pathway (previously using prucalopride) is most effective when done in the pre-operative period (29). Stimulation can also be performed on multiple occasions rather than a single intra-operative episode, which increases the opportunity for intervention. A schedule of five days is used in the present study as this represents the time period during which most patients are expected to regain bowel function (2).

Reviewer 1; Comment 18:

Patient and public involvement. It is stated that a patient advisory group will be convened 6 times across 3 years. How will this be possible if the study proposed is only 2 years?

Thank you. It is important that patients are involved in the set up of the study. As such, the advisory group was convened twice during the year preceding study recruitment, and will be convened twice during the two subsequent years. The following clarification has been made:

— Patient & Public Involvement: A patient advisory group comprising of 4-6 patients with experience of the intervention will be convened on six occasions across three years (including one year prior to study recruitment and two years during recruitment).

Reviewer 1; Comment 19:

In secondary outcomes, Inflammatory response sub-study: peritoneal fluid analysis is not necessarily equivalent to “intestinal inflammation, given that the majority of immune/inflammatory cells of the gut are in the submucosa, whereas the peritoneal cavity has a large influx of peritoneal mononuclear cells. Further, peritoneal fluid cytokines are likely originating from multiple sites including disrupted serosal surfaces at sites of bowel resection/anastomosis and peritoneal membranes at site of abdominal incisions. Perhaps the authors may consider “peritoneal inflammation” vs “intestinal” for clarity and accuracy.

Thank you. We have clarified the approach to collecting these samples and terminology:

— Inflammatory response sub-study: Peritoneal lavage fluid will also be collected at the start and end of surgery. Lavages will be performed by instilling 20ml of sterile 0.9% NaCl onto the small bowel using a foley catheter and bladder tip syringe, followed by collection of 5-10ml of fluid after 30 seconds, as described previously (22). Serum (systemic) and peritoneal lavage (intestinal) levels of tumour necrosis factor (TNF) alpha, interleukin (IL) 1-beta, and IL-6 will be analysed using enzyme-linked immunosorbent assays (ELISA) (23, 24).

— Refs: The FO. Bennink RJ. Ankum WM. et al. Intestinal handling-induced mast cell activation and inflammation in human postoperative ileus. Gut 2008;57:33-40.

Reviewer 1; Comment 20:

SPIRIT checklist. Interventions, points 11b and 11c are missing. For example, subject adherence and compliance to nVNS in the pre-op period can be improved by having research staff check in on day 2-3 via telephone to assess compliance and clarify concerns/questions.

Thank you. We indicated on the submitted SPIRIT checklist that these items were not applicable. It is intended that participants will self-administer the device with no further prompting beyond the training programme. This aligns with the principle of enabling patients to take an active role in their recovery. An objective of this feasibility study is to measure compliance to the stimulation schedule on this basis and to explore barriers/facilitators of compliance during the qualitative sub-study. As such, no protocols aiming to modify compliance during the study are applicable.

Reviewer 2; Comment 1:

Will patients planned to have a diverting stoma, for example to cover a high-risk rectal anastomosis, be eligible for inclusion in the study?

Thank you. Patients who are planned for a diverting stoma are not eligible for the study. In instances where an unplanned stoma is formed, patients remain in the study and are analysed as part of the intention-to-treat population. The following has been clarified:

— Study Eligibility: Planned to undergo elective, minimally-invasive (laparoscopic or robotic) surgery with a planned anastomosis/no routine plans for a diverting stoma, including one of the procedures listed in Table 1.

— Statistical Analysis Plan: Possible reasons for exclusion from the per-protocol population include non-compliance to the device schedule (defined as any deviation from the five-day pre- and post-operative stimulation schedule), conversion from laparoscopic to open surgery, formation of unplanned stoma, return to theatre prior to the return of bowel function, and randomisation errors.

Reviewer 2; Comment 2:

What evidence is available demonstrating that the 0.1Hz biphasic direct current used as a sham stimulation is sub-therapeutic for the purposes of nVNS? This statement should be referenced.

Thank you. In response to a similar comment (Review 1; Comment 10), the following text was added:

— Discussion: Finally, identifying an adequate sham intervention by which to compare nVNS is challenging. In previously healthy volunteer studies, nVNS has been shown to significantly reduce the expression of inflammatory cytokines from baseline but not when using the sham device (32). It is possible that manual instrumentation of the device may be sufficient to elicit some vagal activity, but this is considered minimal and is comparable across both device types.

The appropriate reference is:

— Reference 32: Lerman I. Hauger R. Sorkin L. Proudfoot J. Davis B. et al. Noninvasive Transcutaneous Vagus Nerve Stimulation Decreases Whole Blood Culture-Derived Cytokines and Chemokines: A Randomized, Blinded, Healthy Control Pilot Trial. *Neuromodulation* 2016;19:283-90 PubMed .

Reviewer 2; Comment 3:

Have the authors considered how the crossover design may affect the assessment of blinding at the POD5 time point? i.e. Patients may be adequately blinded pre-operatively but then become unblinded during the post-operative period if the change between active/sham stimulation alters the perceptible stimulus of stimulation? It may be beneficial to assess blinding pre-operatively on the day of surgery as well as on POD5 – particularly given adequacy of blinding is a part of the progression criteria.

Thank you. We agree, this is a very reasonable approach. Since the study is a four-arm, parallel-group study, the performance of blinding will be assessed across all four permutations of device allocation. As such, the possibility of unblinding in some groups (but not in others) will be explored by the blinding poll/qualitative sub-study and this will contribute to the feasibility assessment.

Reviewer 2; Comment 4:

How will non-compliance be defined for the purposes of exclusion of patients from the intention-to-treat analysis?

Thank you, participants who comply fully to the stimulation schedule (i.e. twice daily stimulation for five days before surgery and five days after) will be considered in the per-protocol population. Non-compliance is defined as any deviation from this, and where applicable, participants will be considered in the intention-to-treat population. This has been clarified:

— Statistical Analysis Plan: Possible reasons for exclusion from the per-protocol population include non-compliance to the device schedule (defined as any deviation from the five-day pre- and

post-operative stimulation schedule), conversion from laparoscopic to open surgery, formation of unplanned stoma, return to theatre prior to the return of bowel function, and randomisation errors

Reviewer 2; Comment 5:

Is there any anticipated bias in the MRE sub-study as to whether patients taking part in this will be able to tolerate sufficient oral intake for the enterography oral preparation?

Thank you – this is a very good point. The oral preparation involved in these MR examinations involves a single glass of water. Our previous pilot examinations (currently unpublished) indicate that this is well tolerated and the examination results are meaningful.

— Trial sub-studies; bowel motility sub-study: Oral preparation with a single glass of water will take place 30 minutes prior to the scan.

Reviewer 2; Comment 6:

“Systematic” in paragraph 3 on Page 15 should read “systemic”.

Thank you, this has been corrected.

Reviewer 2; Comment 7:

What data will be collected from patients at the 30-day point as part of the follow-up CRF? The clinical outcomes listed on Page 12 are likely to all be met either as an inpatient or very early following discharge.

Thank you. The occurrence of complications will be collected at the 30-day time point. This will capture events which may occur during the index hospital admission and events which may occur later and lead to re-admission/treatment. The following text has been added:

— Participant timeline: Follow up data (including the occurrence of complications and serious complications) will be collected 30 days after surgery to capture events which may occur during the index hospital admission or after discharge.

Reviewer 2; Comment 8:

Much of the evidence for VNS in abdominal surgery has been in the setting of foregut surgery. This may not be directly applicable to the colorectal setting where the hindgut does not receive direct vagal innervation, and this is a potential limitation of this approach which the authors may like to consider in the discussion.

Thank you. Whilst colonic dysmotility is likely implicated in the pathogenesis of ileus, most previous evidence implicates dysmotility of the small bowel (References: 11, 12, 13, 14). The small bowel is readily innervated by the vagus nerve. The vagal innervation of the colon is debated, but there is general consensus that this receives vagal innervation at least up to and including the splenic flexure (Reference 13).

— Discussion: Its efficacy has been demonstrated consistently in pre-clinical studies, where stimulation of the vagus nerve has been shown to reduce intestinal inflammation and restore small bowel transit through activation of a cholinergic anti-inflammatory pathway (11).

Reviewer 2; Comment 9:

Good luck with the trial, we will be awaiting the results with great interest.

Thank you. We really appreciate your comments.

VERSION 2 – REVIEW

REVIEWER	Gottfried Blackmore, Andres Stanford University, Medicine, Gastroenterology
REVIEW RETURNED	09-Jan-2021

GENERAL COMMENTS	The authors have addressed all of this reviewer's concerns, which for the most part, were due to lack of clarity on behalf of this reviewer towards the nature of this feasibility study vs a pilot clinical trial. Only one remaining minor point of clarification is recommended from the initial review (Reviewer 1; Comment 19) regarding the terminology of "intestinal" inflammation/levels of inflammatory markers. In the revised text: "Serum (systemic) and peritoneal lavage (intestinal) levels of tumour necrosis factor (TNF) alpha, interleukin (IL) 1-beta, and IL-6 will be analysed using enzyme-linked immunosorbent assays (ELISA)..." I would still argue that peritoneal lavage fluid markers reflect "abdominal" levels of inflammatory markers, and not "intestinal". The reference provided (Bennik et al, GUT 2008) is appreciated, and shows that laparotomy (vs laparoscopic/transvaginal approach) is associated with increased inflammation. It does not demonstrate that inflammatory products are originated from the bowel/intestine (as laparoscopic surgery with manipulation of bowel did not yield inflammatory lavage products, not were the laparotomy samples controlled appropriately (manipulated vs non-manipulated). Rather, their observation is a reflection of surgery time, not intestinal source of inflammatory products. Also wish you luck and looking forward to the outcomes from your trials with nVNS.
---

REVIEWER	O'Grady, Gregory University of Auckland
REVIEW RETURNED	06-Jan-2021

GENERAL COMMENTS	The authors have satisfactorily addressed all of our comments and queries. All the best with the study, we look forward to seeing the final results.
--

VERSION 2 – AUTHOR RESPONSE

Reviewer 1; Comment 19:

Regarding the terminology of "intestinal" inflammation/levels of inflammatory markers.

In the revised text: "Serum (systemic) and peritoneal lavage (intestinal) levels of tumour necrosis factor (TNF) alpha, interleukin (IL) 1-beta, and IL-6 will be analysed using enzyme-linked immunosorbent assays (ELISA)...". I would still argue that peritoneal lavage fluid markers reflect "abdominal" levels of inflammatory markers, and not "intestinal". The reference provided (Bennik et al, GUT 2008) is appreciated, and shows that laparotomy (vs laparoscopic/transvaginal approach) is associated with increased inflammation. It does not demonstrate that inflammatory products are originated from the bowel/intestine (as laparoscopic surgery with manipulation of bowel did not yield inflammatory lavage products, not were the lparotomy samples controlled appropriately (manipulated vs non-manipulated). Rather, their observation is a reflection of surgery time, not intestinal source of inflammatory products.

Thank you for the opportunity to correct the terminology. We have amended this section with the text below:

— The inflammatory response sub-study will explore the effect of nVNS on measures of systematic and peritoneal inflammation. This will explore differences in efficacy between randomised arms, which will assist with the design of the final intervention. Recruitment to this sub-study will take place at the lead site only (St. James's University Hospital). Additional eligibility criteria will apply, as outlined in Table 4. Blood draws will be performed at baseline, 2 hours, 24 hours, and 48 hours after the start of surgery. Peritoneal lavage fluid will also be collected at the start and end of surgery. Lavages will be performed by instilling 20ml of sterile 0.9% NaCl onto the small bowel using a foley catheter and bladder tip syringe, followed by collection of 5-10ml of fluid after 30 seconds, as described previously (22). Serum (systemic) and fluid (peritoneal) levels of tumour necrosis factor (TNF) alpha, interleukin (IL) 1-beta, and IL-6 will be analysed using enzyme-linked immunosorbent assays (ELISA).

Other Revisions

In addition to the reviewer comment, we have made a small number of typographical adjustments to improve the flow and readability of the text. These are indicated in red text throughout the marked manuscript.